# Alternatively Spliced Isoforms of the P2X7 Receptor: Structure, Function and Disease Associations

**DOI:** 10.3390/ijms23158174

**Published:** 2022-07-25

**Authors:** Sophie K. F. De Salis, Lanxin Li, Zheng Chen, Kam Wa Lam, Kristen K. Skarratt, Thomas Balle, Stephen J. Fuller

**Affiliations:** 1Sydney Pharmacy School, Faculty of Medicine and Health, The University of Sydney, Sydney, NSW 2006, Australia; sode6437@uni.sydney.edu.au (S.K.F.D.S.); jake.chen@sydney.edu.au (Z.C.); thomas.balle@sydney.edu.au (T.B.); 2Sydney Medical School Nepean, Faculty of Medicine and Health, The University of Sydney, Nepean Hospital, Penrith, NSW 2750, Australia; lali3292@uni.sydney.edu.au (L.L.); klam2624@uni.sydney.edu.au (K.W.L.); kristy.skarratt@sydney.edu.au (K.K.S.); 3Brain and Mind Centre, The University of Sydney, Camperdown, NSW 2050, Australia

**Keywords:** P2X7R, alternative splicing, ATP, cancer, inflammation

## Abstract

The P2X7 receptor (P2X7R) is an ATP-gated membrane ion channel that is expressed by multiple cell types. Following activation by extracellular ATP, the P2X7R mediates a broad range of cellular responses including cytokine and chemokine release, cell survival and differentiation, the activation of transcription factors, and apoptosis. The P2X7R is made up of three P2X7 subunits that contain specific domains essential for the receptor’s varied functions. Alternative splicing produces P2X7 isoforms that exclude one or more of these domains and assemble in combinations that alter P2X7R function. The modification of the structure and function of the P2X7R may adversely affect cellular responses to carcinogens and pathogens, and alternatively spliced (AS) P2X7 isoforms have been associated with several cancers. This review summarizes recent advances in understanding the structure and function of AS P2X7 isoforms and their associations with cancer and potential role in modulating the inflammatory response.

## 1. Introduction

P2X receptors are an ancient family of proteins that are expressed in both primitive and advanced life-forms, including protozoa, algae, flatworms, fish, birds, and mammals [1,2,3,4,5]. They are purinergic membrane receptors that assemble as cation channels following the binding of extracellular adenosine 5′-triphosphate (ATP) [1]. P2X receptors are classified into seven subtypes, P2X1 to P2X7, that are each translated from different genes. P2X7 receptors (P2X7R), encoded by the gene *P2RX7*, are unique from the other subtypes in both their structure and function [2,6].

In mammalian cells, P2X7Rs are widely expressed, with high expression in haematopoietic cells including monocyte/macrophages, dendritic cells, mast cells, lymphocytes, eosinophils, and basophils [7]. Expression in such a diverse array of life-forms and tissues suggests that these receptors have a fundamental role in normal cell physiology and homeostasis. The importance of the P2X7R in normal health and development can be inferred from the phenotypic features of the three lines of *P2rx7* knock-out mice that have been generated [8,9,10]. These mice strains recorded lower levels of interleukin (IL)-1β [8,10], reduced sensitivity to inflammatory and neuropathic pain, and one had an osteoporotic phenotype [9,11].

A unique property of P2X7Rs is the modulation of their activity based on ATP concentration. At low ATP concentrations the receptor forms a cation channel. At high ATP concentrations the receptor forms a non-selective pore that allows molecules up to 900 Da to pass through the membrane, consequently inducing cell apoptosis [12]. Many other cellular responses are triggered when ATP binds to the receptor including cytokine secretion, the shedding of cell surface molecules, cell proliferation, and the attenuation of P2X7R-dependent phagocytosis [13,14,15,16]. The mechanisms by which the receptor engages in such a diverse array of responses is unknown. It has been hypothesised that alternative splicing, which allows one gene to produce multiple isoforms, enables alternatively spliced (AS) P2X7 isoforms to form receptors that engage in variable cellular roles by altering P2X7R structure and function. This review will provide a detailed assessment of our current understanding of the effects of the alternative splicing of *P2RX7* messenger ribonucleic acid (mRNA) on the receptor’s structure, function, and disease associations. In addition, the reader is referred to other recent reviews on the P2X7R [16,17].

## 2. Structure of the P2X7R

Prior to 2019, preliminary models of the P2X7R were developed using x-ray crystallography and homology modelling, but these methods were limited by their inability to define the structure of intracellular domains of the receptor [18,19,20,21]. In 2019, McCarthy et al. developed cryogenic electron microscopic structures of the open and closed states of the rat P2X7R. These structures provided the first insight into the complete P2X7R structure, including the intracellular domains and ATP-binding sites (Figure 1A,B) [22].

Each P2X7 subunit that forms the trimeric P2X7R is a product of the human *P2RX7* gene, located on chromosome 12 [24]. The full-length P2X7A subunit, comprised of 595 amino acids, is translated from a 13 exon mRNA transcript [24]. P2X7A has an extracellular domain located between two transmembrane domains (TM1 and TM2), with intracellular amino and carboxy termini (Figure 2A) [20,25,26]. Three P2X7 subunits assemble to form the P2X7R channel which is lined by TM2 segments that are responsible for channel opening and ion selectivity [27,28,29]. Three ATP-binding sites are located at the interface between adjacent P2X7A subunits (Figure 1B) [30]. Brief exposure to extracellular ATP, released from several cellular sources in response to cell stress and tissue damage, induces the rapid opening of a cation channel. Ca^2+^ and Na^+^ moves into the cell while K^+^ moves out of the cell [31]. Repeated or prolonged ATP activation results in an irreversible increase in membrane permeability by forming a larger pore, allowing hydrophilic molecules up to 900 Da to pass through the cell membrane [31,32,33]. However, patch-clamp electrophysiology studies performed by Riedel et al. have shown that single channel kinetics and permeation properties do not change during prolonged receptor activation, challenging the idea of pore dilation [34]. Li et al. found that the characteristic shift in equilibrium, or reversal potential found with prolonged P2X7R activation, resulted from time-dependent alterations in the concentration of intracellular ions rather than the opening of a larger pore [35]. Furthermore, substituted cysteine accessibility mutagenesis experiments and single channel studies have provided no evidence of pore formation following prolonged ATP activation [36]. Consequently, the existence of other pathways has been postulated to explain permeability to relatively large cations such as NMDG^+^ and ethidium^+^ [37,38,39]. P2X7R activation is a major physiological stimulus for the release of IL-1β, matrix metallopeptidase 9, CD23, tumour necrosis factor (TNF) α, transforming growth factor α, and proteolytic cleavage of soluble IL-6 receptor [40,41,42]. In the absence of extracellular ATP, the P2X7R functions as a macrophage scavenger receptor [43]. As a result of these functions, P2X7Rs have central roles in inflammatory signalling pathways and innate immune responses [17].

## 3. P2X7R Domains

### 3.1. Amino Terminal

Preceding TM1, a short amino terminal region consisting of residues 1 to 27 is encoded by exon 1 (Figure 2A,B). The amino terminal contains a protein kinase C phosphorylation consensus site [TX(K/R)] that is conserved in all P2X subtypes [44]. This site has been proposed to play a role in ATP-mediated receptor activation and desensitisation [45].

### 3.2. Extracellular Domain

The extracellular domain, comprising residues 51 to 333, contains the orthosteric ATP-binding sites [46,47]. The location of conserved residues required for ATP-binding in each of the subunits has been inferred from the agonist-bound crystal structure of the zebrafish P2X4 receptor [20]. The ammonium groups of K64 and K66, located within the first subunit, are encoded by exon 2 and are predicted to form hydrogen bonds with ATP phosphate groups [6,20]. In the adjacent P2X7 subunit, ATP interacts with N292 and R294, which are encoded by exon 8 and K311, which is encoded by exon 9 (Figure 2B) [47,48].

In addition to the ATP-binding sites, crystallisation and mutagenesis studies have identified an allosteric binding pocket for P2X7R antagonists in the extracellular domain, principally mediated by hydrophobic interactions with F95, F103, M105, F293, and V312 [19,49]. In addition, two series of residues that are unique to the P2X7R, a loop insertion (residues 73 to 79) and residues T90 and T94, have been associated with potent antagonist binding [50].

### 3.3. Carboxy Terminal

Compared to other P2X receptor family members, the P2X7R has a unique, long cytoplasmic carboxy terminal. In addition to the formation of the large conductance pore [51,52], the carboxy tail is critical for signalling that it is independent of channel or pore formation [51,53,54]. Several lipid and protein binding motifs that potentially interact with adaptor and/or effector proteins have been identified in this domain that extends from residues 357 to 595 (Figure 2C) [55]. A P-X-X-P motif has been identified at residues 441 to 460. This cellular sarcoma tyrosine kinase homology 3 (SH3) binding epitope allows the carboxy terminal to bind to signalling proteins that contain SH3 domains [56]. P2X7Rs have therefore been linked to processes associated with the SH3 domain including cell growth regulation, phagocytosis, and cytoskeleton control [43,57,58].

Residues 436 to 531 have sequence similarity to a segment of the TNF receptor 1 (TNFR1), including the death domain [55]. The TNFR1 death domain located on the P2X7R may interact with the adaptor molecule, the TNF receptor-associated death domain, which is the first step in signal transduction leading to activation of apoptosis-mediating caspases [59,60]. Consequently, P2X7R-induced apoptosis may be stimulated by several cell death pathways, including the efflux of K^+^ ions through the P2X7R pore that results in c-Jun N-terminal kinase phosphorylation [61], and pore-independent signal transduction via the carboxy terminal TNFR1-associated death domain [62].

The carboxy-terminal is important for receptor trafficking and binding to cytoskeletal proteins. These processes have been attributed to the trafficking domain from residue 551 to 581, the actin filament binding domain spanning from residue 389 to 405 that has homology to *Mycoplasma genitalium* cytoadherence high molecular weight protein 3, and a membrane protein—cytoskeleton linking domain from residue 494 to 508 that is homologous to the ankyrin containing protein C18H2.1, which is involved in linking membrane proteins to the cytoskeleton [55,63,64]. The long carboxy terminal is therefore critical to facilitating localization of the receptor.

The P2X7R carboxy terminal has a lipopolysaccharide (LPS) domain that spans from residues 573 to 590, resembling the LPS-binding region of serum LPS-binding proteins. This domain is responsible for neutralising LPS-associated endotoxic activity [65].

In addition to domain-specific functions that are distinct from P2X7R pore activity, the carboxy terminal contains a cytoplasmic cysteine rich (C rich) domain and a cytoplasmic ballast that are required for P2X7R pore formation [51]. Palmitoylation of the C rich domain modulates the receptor function by binding cysteine residues to cholesterol in the surface membrane, enabling interactions between the P2X7R and membrane phospholipid rafts [22,66,67,68]. These cysteines are encoded by exon 13, and in the rat P2X7R consist of C residues 477, 479, 482, 498, 499, 506, 572 and 573, with another C552 residue in the human and mouse P2X7R [22]. The attachment to the lipid raft is a key regulator of P2X7R desensitisation [22] and channel opening [67,68]. The last 120 amino acids of the carboxy terminal make up the cytoplasmic ballast. This ballast has been implicated in the P2X7R’s pore dilation and initiating cytolytic signal transduction, likely via the dinuclear zinc ion complex and high-affinity guanosine nucleotide binding site located within the ballast [22,69,70].

## 4. Alternative Splicing

Alternative splicing is a universal regulator of gene expression that from a single gene generates various mRNAs which differ in their untranslated regions or protein coding sequences. Alternative splicing mechanisms include skipping exons (the removal of cassette exons), retaining introns, selecting mutually exclusive exons, and using alternative 5′ and 3′ splice sites that affect the boundaries between introns and exons, contributing to transcript diversity. A functional protein may not result from alternative splicing for several reasons including the generation of a non-coding transcript, unstable RNA, the alteration of RNA localisation that prevents translation, or the generation of a non-coding RNA that competes with regulators of coding RNAs [71]. There is increasing evidence that alternative splicing can affect all aspects of ligand-gated ion channel function including channel gating, ion sensitivity, and ligand binding [72].

Alternative splicing is common, and the use of genome-wide scanning is found in over 95% of human multi-exon genes [73,74] The relative abundance of different AS isoforms is determined by several factors including splice site strength, cis-regulatory sequences in pre-mRNAs, the expression of trans-acting factors such as RNA binding proteins and splicing factors, and the alteration of cis-regulation splicing [71]. Cis-regulation of splicing is altered by inherited genetic variations including single nucleotide polymorphisms (SNPs) due to high sequence sensitivity [75].

Variations in P2X7R function between human subjects can therefore be explained by both the inheritance of SNPs [76,77] and the formation of AS isoforms [6,69,78,79]. To date there are over 20 *P2RX7* mRNA isoforms listed in the National Center for Biotechnology Information (NCBI) and Ensembl databases (Table 1). There is no standardised nomenclature for *P2RX7* mRNA or P2X7 AS isoforms. The NCBI database lists isoforms as variant numbers and Ensembl as P2X7 numbers, which do not correspond to the alphabetic order originally used by Cheewatrakoolpong et al. [69]. However, because the alphabetic nomenclature is commonly referred to in the literature, we have aligned the NCBI and Ensembl transcript numbers to the corresponding alphabetised isoform (Table 1). There are eight P2X7 protein AS isoforms listed by UniProtKB, although many other mRNA isoforms may be translated if alternative start codons are used (Table 1 and Figure 3). Of the known *P2RX7* mRNA AS isoforms, we and others have found five that translate to proteins in cell model systems [6,69,79]. *P2RX7A* is a full-length mRNA that contains 13 exons and no introns (Figure 3) [69]. The protein translated from *P2RX7A* mRNA is the canonical form described earlier in the review, comprising an intracellular amino terminal followed by a TM1, extracellular domain, TM2, and a carboxy terminal [69]. *P2RX7B* transcribes the intron located between exon 10 and 11 in the deoxyribonucleic acid (DNA) sequence that introduces a premature stop codon [69]. The resulting P2X7B protein lacks the carboxy terminal encoded by exons 11, 12 and 13, and has 18 alternate amino acids inserted after residue 346 [69]. As well as the inclusion of intron 10, *P2RX7E* mRNA skips exons 7 and 8 and is translated to a protein with a truncated carboxy terminal and lacks the critical residues required for ATP binding [69]. *P2RX7J* is translated to a 258 amino acid protein that lacks part of the extracellular domain encoded by exons 9 and 10, the entire TM2 encoded by exons 10 and 11, and the carboxy terminal encoded by exons 11, 12 and 13, acquiring 10 unique residues in its new carboxy terminal domain [79]. *P2RX7L,* like *P2RX7E*, lacks exons 7 and 8 that encode part of the extracellular domain containing the ATP binding site [6].

**Figure 3 ijms-23-08174-f003:**
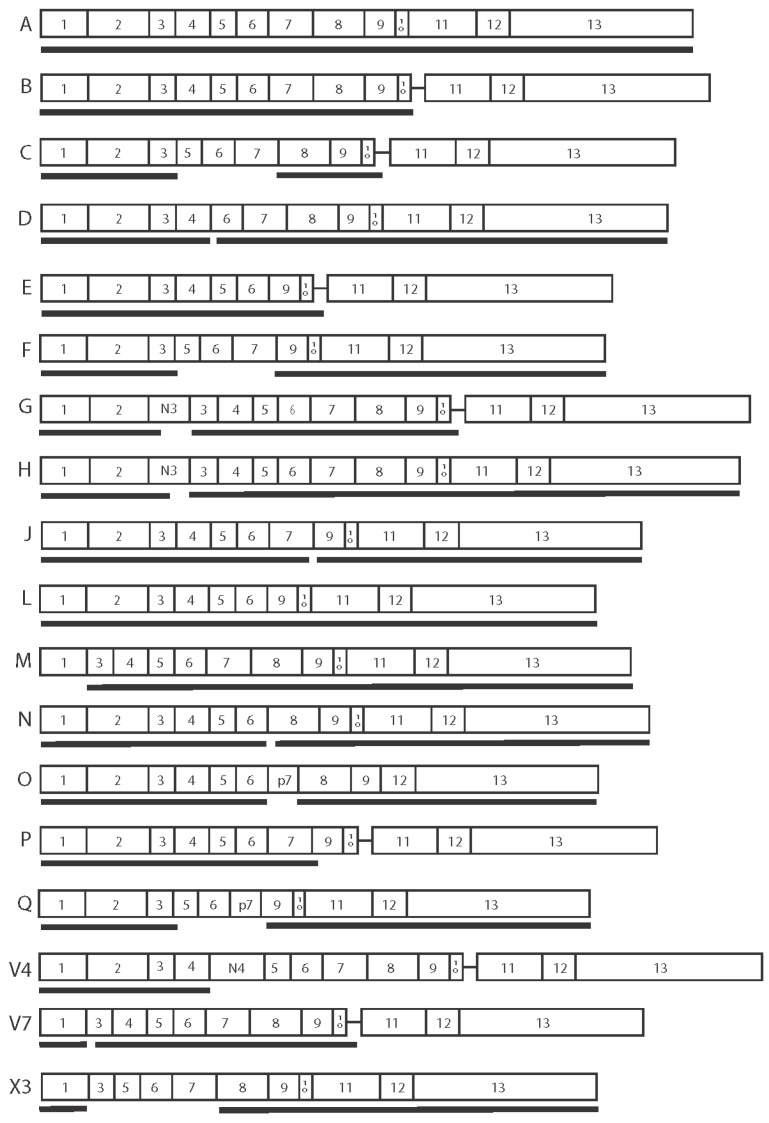
*P2RX7* mRNA isoforms and predicted proteins. Diagrammatic representation of the exons (open numbered boxes) comprising each transcript (mRNA variant names are listed on the left of each transcript). The thin line joining exon 10 to exon 11 represents the retained intron 10. N3 and N4 are alternative exons; p7 indicates that part of the sequence for exon 7 is missing. Thick black lines represent the proteins that could be translated beginning at the primary start codon in exon 1 or at an alternative site.

*P2RX7H* (also referred to as *P2RX7-V3*) is also a key AS isoform that has been associated with cancer. It is a long non-coding RNA (lncRNA) that contains an extra exon referred to as N3 [82]. LncRNAs are recognised as key regulators of oncogenic pathways [83], and are involved in proliferation, replicative immortality, resistance to growth suppressors, angiogenesis, resistance to apoptosis, and cancer metastasis [84].

The P2X7K AS isoform has been found in mice (mP2X7K) and rats. This isoform escapes deletion in the GlaxoSmithKline P2X7R-deficient mouse line, which was generated by inserting a lacZ/Neo reporter cassette into exon 1 of *P2rx7* [9]. This occurs because *P2rx7k* has an alternative exon 1 located in the intronic region between exons 1 and 2 of the *P2rx7* gene that replaces the first 42 amino acid residues of P2X7A with 39 different amino acid residues. An alternative amino terminal and TM1 are translated in the mP2X7K protein [81]. A second *P2rx7* gene deletion mouse has been generated by Pfizer by inserting a neomycin cassette into exon 13 [8]. Two AS isoforms, mP2X7 13b and mP2X7 13c, which are not found in humans, escape disruption in the Pfizer mouse. Like the human P2X7B AS isoform, mP2X7 13b and mP2X7 13c have alternate carboxy termini due to alterations in exon 13. The first AS isoform, mP2X7 13b, terminates at T431 and the second, mP2X7 13c, has an extra 11 amino acids in the carboxy terminal [85].

To date, P2X7A has been found to form heterotrimeric receptors with the P2X7B, P2X7J, and P2X7L subunits, and mP2X7K with mP2X7A [6,52,79,81]. Heterotrimeric P2X7Rs composed of different combinations of AS isoform subunits have been found to affect several functions of the P2X7R. In vitro, P2X7B potentiates receptor function when co-assembled with full-length P2X7A (Figure 4) and transfected into human embryonic kidney-293 (HEK293) cells [52]. The deletion of the long carboxy terminal in P2X7B appears to enable the receptor to retain the growth-promoting activities of P2X7A, stimulating cell proliferation in response to ATP while losing the cytotoxicity that is related to pore formation [52]. When co-expressed with P2X7A in HEK293 and Madin-Darby canine kidney cell lines, P2X7J acts as a dominant-negative isoform on P2X7A [79]. It forms a non-functional P2X7A/P2X7J heterotrimer that is trafficked to the plasma membrane but fails to undergo apoptosis [79]. mP2X7A/mP2X7K receptors are functional [86]; however, as P2X7K AS isoforms have only been identified in rodents to date, they will not be discussed further. P2X7A/P2X7L receptors are functional but have reduced pore formation, likely because P2X7L lacks an ATP-binding site [6].

## 5. Inherited Variation in *P2RX7* mRNA Splice Sites

The contribution of inherited genetic variation to the diversity of mRNA AS isoforms is well established, and variations in pre-mRNA sequences can affect several cis-acting regions that control alternative splicing [87]. SNPs that occur at the highly conserved donor and acceptor di-nucleotides predictably affect splicing, and when these occur near verified exon boundaries, they are annotated in databases such as the NCBI Single Nucleotide Polymorphism Database (dbSNP) [88,89]. However, a large proportion of SNPs also occur at sites where the effect on splicing is less clear, including less conserved sites close to the intron/exon boundaries, the intronic branch-point, and within intronic or exonic splicing enhancer or suppressor sequences [90,91]. The complexity of isoform expression is further increased by allele-specific alternative splicing whereby genetic variations that are present in either allele may be expressed at different levels [92]. Allele-specific alternative splicing can be complete, where one allele encodes one isoform and the other results in the alternative isoform, or partial, where different alleles encode different percentages of more than one isoform [92,93,94].

Our group was the first to describe an inherited splice site mutation (rs35933842, G > T) located at position +1 in the first intron of *P2RX7* that resulted in a P2X7 protein null allele in 1–2% of the Caucasian population [80]. Disruption of the guanosine-5′ donor splice site with this SNP prevents splicing out of intron 1. The corresponding transcript likely becomes a target for nonsense-mediated mRNA decay since a premature stop codon is located a short distance within the included intron sequence. Since the inheritance of rs35933842 generates a null allele, only one allele can be translated to a protein, meaning that subjects who are heterozygous for function altering SNPs become functionally homozygous [80].

Inherited, allele-specific alternative splicing likely contributes to the significant variability in the patterns of AS isoform mRNA expression observed between individuals. We recently studied the association between a haplotype block (rs208307, rs208306, rs36144845, rs208308, rs208309, and rs275655596) tagged by rs208307 (641-5 C > G), which is located at an acceptor splice site in intron 6, and the expression of exon 7 and exon 8 deletion (EX7_EX8del) mRNA transcripts in normal human subjects [6]. Consistent with allele-specific alternative splicing, the levels of EX7_EX8del mRNA were found to be different depending on whether subjects were homozygous or heterozygous for the minor allele at rs208307. The mean allele frequency for the rs208307 SNP was 0 in subjects who did not express the EX7_EX8del mRNA. Comparatively, the mean allele frequency for those who expressed EX7_EX8del mRNA was 0.5 (*p* < 0.0001). The association between the EX7_EX8del mRNA and the haplotype can be explained by exon skipping that was reported in vitro in P2X7-transfected HEK293 cells when this haplotype was present [6]. Allele-specific alternative splicing, therefore, likely plays a role in the genetic regulation of the mRNA levels of the two known EX7_EX8del AS isoforms, P2X7E and P2X7L. It is anticipated that SNPs in other cis-acting regions that control alternative splicing affect the mRNA levels of other AS isoforms.

It is now known that the vast majority of trait- or disease-associated nucleotide variant loci are in intronic and intergenic non-coding regions [95]. Intronic variants mainly regulate biological activities by affecting mRNA splicing [96], and at present there are over 400 *P2RX7* intronic SNPs (minor allele frequency >0.005) entered in the dbSNP database [97].

## 6. Role of P2X7R Isoforms in Cancer

Following the discovery and characterisation of P2X7R AS isoforms that interact with full-length P2X7A subunits and alter receptor function, there has been an interest in the trophic effects of these isoforms and cancer cell proliferation. Extracellular ATP is increased in the tumour microenvironment [98], and, consequently, an associated increase in P2X7R activation is expected. It has been proposed, based on structure and associated function, that P2X7B and P2X7H isoforms may predispose to cancer. P2X7B has a truncated carboxy termini, abolishing its ability to induce cell apoptosis, ultimately resulting in cell proliferation. P2X7H acts as a lncRNA, regulating oncogenic pathways as described earlier in the review [82,83].

### 6.1. Adenocarcinoma of the Lung

Lung cancers are divided into small-cell carcinoma and non-small-cell carcinoma based on histology, prognosis, and response to specific therapies [99]. Adenocarcinoma is the most common type of non-small-cell-carcinoma and is comprised of malignant epithelial cells with glandular differentiation [99]. P2X7R expression has been detected in adenocarcinoma and squamous cell carcinoma of the lung with one study finding no difference in expression between the two histological subtypes [100]. Benzaquen et al. found that lung adenocarcinoma cells expressed P2X7A, P2X7B, and additional bands from 56 to 80 kDa that could correspond to other AS isoforms when analysed using western blot and the L4 monoclonal antibody that binds to the extracellular domain of P2X7Rs [101]. Immune cells that were positive for leukocyte common antigen (CD45+) in the tumour microenvironment had lower levels of P2X7R pore function compared to CD45+ cells outside the tumour area. This difference in function may be explained by higher levels of P2X7B mRNA in immune cells in the tumour microenvironment compared to normal lung tissue [101].

### 6.2. Neuroblastoma

Neuroblastomas arise from neural crest cells located in the adrenal medulla or sympathetic chain. They are the most common extracranial cancers in childhood and patients are divided into low, intermediate, and high-risk. High-risk patients make up half of newly diagnosed patients and have poor survival outcomes (60% of patients have a survival duration of five years) compared to low risk patients (95% of patients have a survival duration of more than five years) [102].

Neuroblastomas commonly metastasise to the bone marrow. Ulrich et al. found that bradykinin contributes to metastasis by increasing the sensitivity of C-X-C chemokine receptor type 4 to stromal cell-derived factor-1, increasing the sensitivity of P2X7Rs to ATP, and increasing P2X7B expression to a greater extent than P2X7A [103]. Bradykinin induced P2X7B isoform expression would allow neuroblastoma cells to use high extracellular ATP concentrations in the bone marrow as a growth and seeding stimulus while resisting apoptosis [103].

### 6.3. Osteosarcoma

In the bone, P2X7Rs are expressed by osteoclasts, osteoblasts, and osteocytes [104,105,106]. Osteosarcomas are the most common form of bone cancer and likely arise from transformed multipotent mesenchymal stem cells that develop into several bone differentiation lineages [107]. Clinical outcomes are generally poor and are related to complex genetic and epigenetic factors [108]. A recent study suggested that P2X7B expression in osteosarcoma cells greatly enhanced tumour growth and favoured metastasis [109]. However, the application of P2X7B antagonists has not been associated with the expected tumour suppressive effects [110]. Differences in results between studies may be explained by the different cell lines used and an inadequate representation of tumour microenvironments, triggering inconsistent expression of P2X7B in different experiments. As the expression of P2X7R isoforms are highly flexible, changes in isoform expression patterns between individuals requires further research.

### 6.4. Cervical Cancer

Cervical cancer is one of the most common cancers diagnosed in women [111], and virtually all cases are caused by infection with human papillomavirus (HPV) [112], such that screening for HPV subtype 16 oncoprotein has a higher sensitivity than cytological testing [113]. HPV contributes to the development and progression of cervical cancer by disrupting alternative splicing [114]. P2X7J has been identified in human cervical squamous cell carcinoma cells and normal cervical cells, although it is likely widely expressed [79]. Feng et al. observed the higher expression of P2X7J in cancerous compared to healthy cells. When observing the expression of P2X7A compared to P2X7J, the higher expression of P2X7J was found in cervical cell carcinoma cells, with the ratio reversed in normal cervical cells [79]. As a non-functional isoform, P2X7J likely contributes to cell proliferation by preventing cell apoptosis that is associated with the unmutated full-length P2X7A. Overall, HPV and P2X7J have been independently associated with cervical cancer, but it would be interesting to also study an association between HPV and P2X7J expression.

### 6.5. Melanoma

A melanoma is a skin cancer that affects melanocytes, a type of cell that originates from neural crest cells [115]. Native human melanoma cells and cell lines have been found to express functional P2X7R that, when exposed to an agonist, induced apoptosis [116,117]. AS isoforms, however, have only been studied in uveal melanoma where an association was found between uveal melanoma and the AS isoform *P2X7RH* that acts as a lncRNA [82]. LncRNAs have no protein-coding potential but are implicated in several cellular processes with distinct regulatory roles, including tumorigenesis [118,119]. Pan et al. reported high levels of *P2RX7H* (Table 1) mRNA in uveal melanoma cell lines, and the downregulation of *P2RX7H* inhibited cell migration and colony formation [82]. Furthermore, in a xenograft model in nude mice, stable knockdown using *P2RX7H* short hairpin RNA (shRNA) in the metastatic uveal melanoma cell line, MUM2B, suppressed tumour growth [82]. These studies show that P*2RX7H* likely has a role in uveal tumour progression and is potentially an informative biomarker and therapeutic target.

### 6.6. Glioblastoma Multiforme

Glioblastoma multiforme is a rare type of brain cancer that most commonly presents after the age of 65 [120]. It arises from a complex interplay of genes that results in the proliferation of cells arising from an astrocyte lineage [121]. The standard treatment is surgery followed by radiotherapy and chemotherapy, however treatment failure frequently occurs. The median overall survival is approximately one year and the relative survival rate at five years is 5% [122]. Monif et al. [123] detected P2X7Rs in glioma tumour cells and microglia that surround the glioblastoma tumours [123]. These receptors were able to form a pore that was inhibited by the P2X7R antagonists, oxidised ATP and brilliant blue G (BBG), reducing microglia and glioma cell numbers [123]. Ryu et al. reported similar results, observing the upregulation of P2X7Rs in glioma cells using a C6 glioma murine model of glioblastoma [124]. Intravenous BBG, which crosses the blood brain barrier, inhibited tumour growth. There was no effect on astrogliosis that suggested the effect of BBG on tumour growth was specific to glioma cells rather than an effect on microglia [124]. The results of these two studies seem counterintuitive, as pore activity should logically be associated with cell apoptosis while inhibiting pore activity should result in cell proliferation.

Tamajusuku et al. [125] and Fang et al. [126] reported contrasting results to Monif et al. [123], and Ryu et al. [124], and Tamjusuku et al. [125] found that ATP and BzATP induced apoptosis in GL261 murine glioma cells that was reduced by the stable RNA interference of P2X7Rs and treatment with oxidised ATP [125]. Similarly, Fang et al. [126], using the same C6 glioma cells as Ryu et al. but transplanted into rats rather than mice, reported that P2X7R shRNA or treatment with BBG was associated with tumour growth by directly promoting cell proliferation and angiogenesis [126].

To address these discordant results, Kan et al. [127] in 2020 used the P2X7R antagonist AZ10606120 against the U251 human glioblastoma cell line and native human glioma cells [127]. In this study, there were no reductions in tumour cell numbers with BBG or oxidised ATP in either U251 cells or human glioma samples. However, AZ10606120 significantly reduced U251 and native human glioma tumour cell numbers and was more effective than the conventional chemotherapeutic agent, temozolomide, in U251 cells [127]. The differences between tumour cell numbers when treated with the different P2X7R antagonists (BBG, oxidised ATP and AZ10606120) can be attributed to the higher selectivity and potency of AZ10606120 with a half maximal inhibitory concentration (IC50) of 10 nM, compared to 200 nM for BBG and 100 μM for oxidised ATP [128,129,130].

The differential expression of P2X7R AS isoforms may also contribute to conflicting results from studies in glioblastoma multiforme. Zanoni et al. [131] found that the differential expression of P2X7A and P2X7B affected sensitivity to ATP-induced cytotoxicity in glioma cell lines [131]. High P2X7A-expressing GB40 cells were more sensitive to ATP-induced cytotoxicity compared to low P2X7A-expressing GB48 cells [131]. Irradiation caused pyroptosis, cell death, and DNA damage in GB40 and GB48 cells, but high P2X7A-expressing GB40 cells were more susceptible to ATP-induced cytotoxicity than the GB48 cells. Furthermore, irradiation-selected radiation-resistant clones in both cell lines expressed higher levels of *P2RX7B* mRNA and lower levels of *P2RX7A* mRNA [131]. Zanoni et al. [131] proposed that the irradiation of glioma cells increased the ratio of P2X7B to P2X7A, contributing to radiation resistance and tumour recurrence [131]. The results are promising but further research is required to see whether these findings cross over into in vivo studies and whether P2X7B antagonists would be effective additives to radiation therapy.

### 6.7. Leukaemia

A range of leukaemias express *P2RX7* mRNA, and higher positive rates and relative expression levels have been found in acute myeloid leukaemia (AML), acute lymphoblastic leukaemia, chronic myeloid leukaemia, and myelodysplastic syndrome patients compared to normal donor bone marrow mononuclear cells [132]. In AML, *P2RX7* mRNA levels varied between subtypes, with higher P2X7R expression in monoblastic and erythroid leukaemias compared to less differentiated leukaemias including acute promyelocytic leukaemia [132]. Higher P2X7R expression was further correlated with poorer prognosis and lower response rates after induction chemotherapy in AML [132].

In terms of *P2RX7* AS mRNA in AML, Pegoraro et al. observed higher *P2RX7A* and *P2RX7B* mRNA levels in AML cells compared to myelodysplastic cells, a precursor to AML [133]. Interestingly, *P2RX7B* mRNA levels were higher in patients at relapse, and the use of a remission-inducing therapy, daunorubicin, was associated with higher levels of *P2RX7B* and lower levels of *P2RX7A* mRNA. In cell culture, AML cells were less likely to undergo apoptosis when treated with daunorubicin if *P2RX7B* mRNA levels were high. These data were reproduced in a murine model using a human promyelocytic cell line where treatment with a P2X7R antagonist (AZ10606120) in combination with daunorubicin resulted in a decrease in both *P2RX7A* and *P2RX7B* mRNA and protein levels [133]. The same authors had previously shown that daunorubicin induced ATP release from AML cells [134]. Daunorubicin may therefore differentially mediate pore formation and apoptosis in cells that express high levels of P2X7A, resulting in the selection of apoptosis resistant AML cells that express high levels of P2X7B. Overall, these studies support a role for P2X7R AS isoforms in AML relapse and resistance to chemotherapy, supporting the possibility that using P2X7R antagonists in combination with chemotherapy may overcome the development of resistance.

## 7. Huntington’s Disease

Huntington’s disease is an autosomal dominant, inherited condition that is caused by a mutation in the Huntingtin gene [135]. Normally, exon 1 of the gene has 6 to 35 CAG repeats. In Huntington’s disease, exon 1 has more than 39 CAG repeats [135]. The Huntingtin mutation has been correlated with cell apoptosis in the striatum neurons, resulting in involuntary movements, a decline in cognition, and changes in mood [136]. The P2X7R has been proposed to participate in neuronal apoptosis by increasing the membrane permeability to Ca^2+^ ions [137]. In a study performed by Ollà et al., post-mortem Huntington’s disease brains were found to have higher levels of *P2RX7* mRNA and P2X7R protein compared to controls [136]. When AS isoforms were observed, P2X7A and P2X7B AS isoforms were found to be upregulated in Huntington’s disease. The increase in P2X7A corresponds with the receptor’s ability to induce neuronal apoptosis. The upregulation of P2X7B appears paradoxical at first as this isoform cannot induce cell apoptosis. Ollà et al. proposed that the upregulation of P2X7B may be a protective mechanism against cell apoptosis in Huntington’s disease, or alternatively a result of inflammation [136]. It is important to also consider whether the participants had upregulated P2X7R expression before disease presentation, whether it was the result of the disease, or whether it was the result of taking a medication. Further research is required to support the results from the study.

## 8. Inflammation

Several mechanisms have been proposed for pathophysiological roles of the P2X7R in inflammation. A well-established mechanism is through the P2X7R—NOD-, LRR- and Pyrin domain-containing Protein 3 (NLRP3) inflammasome—proinflammatory cytokine axis [138]. Several studies have demonstrated the early accumulation of ATP following cellular injuries, including via the release of ATP through pannexin-1 following autophagic cell apoptosis, causing activation of P2X7Rs [9]. As a result, K^+^ efflux activates the assembly of the NLRP3 inflammasome complex, leading to the activation of caspase-1 and maturation of IL-1β. The transmembrane movement of other cations resulting from P2X7R activation is also essential for the release of pro-inflammatory cytokines [10]. In human macrophages and immune cells, this inflammatory axis is believed to be part of a two-step model which requires the highly concentrated exogenous ATP and LPS as a priming stimulant [11]. However, evidence has suggested that in primary circulating monocytes, a single stimulant is enough to trigger an endogenous release of ATP for downstream cascades [13]. A less established mechanism is the activation of the NLRP3 inflammasome through the production of reactive oxygen species that is stimulated by a P2X4/P2X7/pannexin-1 complex. Hung et al. found that eliminating any component of the complex abolished NLRP3 inflammasome activation, suggesting that alterations to the complex would influence the immune response [139,140].

Monocytes are bone marrow derived cells that are critical for cellular innate immunity. After detecting infection, blood monocytes migrate into tissues and differentiate into macrophages. Macrophages initially produce a pro-inflammatory response that following pathogen clearance changes to an anti-inflammatory and wound healing response [141]. Monocyte or macrophage receptor binding to micro-organisms elicits phagocytosis or cytokine secretion through changes in gene expression and alternative splicing [142,143]. In vitro, gram-negative bacteria-derived LPS binding to toll-like receptor 4 elicits this inflammatory response in polarized “M1” macrophages that is enhanced by interferon γ [144]. Polarisation of anti-inflammatory “M2” macrophages, which are divided into M2a to M2d subcategories, occurs via IL-4 for tissue repair, allergic and anti-parasitic responses, IL-1β for humoral immunity and T-helper 2 response, or IL-10 for anti-inflammatory response and tissue repair [145]. The ability to polarise monocytes and macrophages in the laboratory using a limited number of cytokines hints at the even greater plasticity of macrophages at sites of inflammation in the body. To study the effect of macrophage polarisation, Pelegrin et al. used an in vitro polarisation gradient of cytokines and endotoxins to generate a five stage M1 through M2 polarisation model [146]. In the early M1 states, intracellular IL-1β concentration was high and all stimuli, except ATP, continued to activate the inflammasome and release IL-1β. In later polarisation states, ATP no longer signalled to caspase-1 activation that was mediated by a direct reduction in reactive oxygen species production and trapping of the inflammasome complex actin cytoskeleton clusters. Pelegrin et al. concluded that uncoupling of P2X7R from caspase-1 activation may be a trigger in the switch from a pro-inflammatory M1 to anti-inflammatory M2 macrophages [146]. Although there are no reports of P2X7R AS isoforms in native cells at inflammatory sites, it is tempting to hypothesise that alternative splicing of key domains of the receptor may also function to modulate the process of inflammation from pro-inflammatory actions to resolution and healing.

## 9. Conclusions

The effect of both allele- and tissue-specific alternative splicing on P2X7R structure and function have important consequences for the roles these receptors play in multiple cell types in health and disease. Furthermore, there are practical implications for the P2X7R as a pharmacological target. The presence of tissue-specific isoforms would allow the design of drugs with specific actions on given physiological processes, while knowledge of allele-specific alternative splicing would assist in predicting responses and reduce the risk of side effects. An understanding of alternative splicing is essential for emerging RNA therapies which use RNA interference methods to silence specific mRNA isoforms.

Patterns of alternative splicing constantly change under different physiological conditions, allowing gene expression to respond and adapt to changes in the environment. Analyses of isoform regulation in inflammatory cells by cytokines and under different environmental conditions may be pivotal in understanding inter-individual differences in innate immunity. It appears that only a small component of the regulation and impact of alternative splicing on the structure and function of the P2X7R have been characterised, and the molecular basis of alterations in receptor function remain to be elucidated. The recent release of highly accurate computational modelling methods such as AlphaFold, RoseTTAFold, and NanoNet are promising tools that can be used to predict the function of different heterotrimeric isoform combinations. The visualisation of the structures alongside molecular dynamic simulations would be beneficial to further understand the role of P2X7R alternative splicing in inter-individual disease variability.

## Figures and Tables

**Figure 1 ijms-23-08174-f001:**
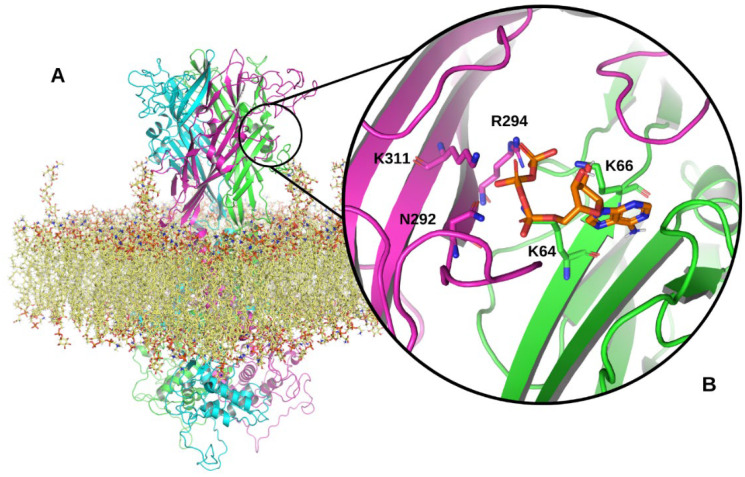
Model of the full-length human P2X7R. (**A**) Structure of the trimeric P2X7R within an epithelial membrane model. The three P2X7 subunits are shown in magenta, cyan and green ribbon representations. The epithelial membrane is shown in yellow. (**B**) A zoomed in view of a P2X7R ATP-binding site located between two P2X7 subunits in the extracellular domain of the receptor. ATP is shown in orange interacting with N292, R294, and K311 on one subunit and K64 and 66 on the adjacent P2X7 subunit. Structural model produced using PyMOL software version 2.5.2, Schrödinger, Inc., New York, NY, USA [23].

**Figure 2 ijms-23-08174-f002:**
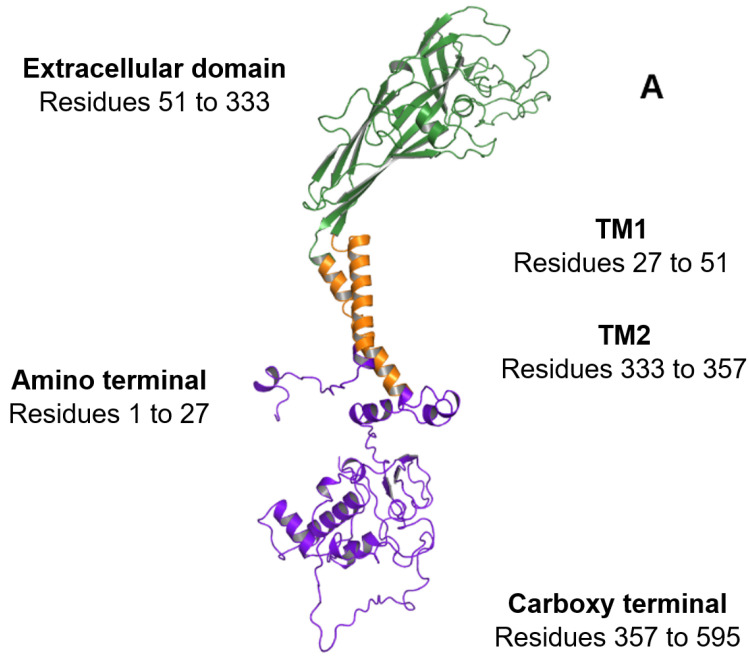
Overview of the regions constituting the P2X7A subunit. (**A**) Ribbon representation of the P2X7A subunit structure. The intracellular amino and carboxy termini are shown in purple, the transmembrane domains (TM1 and TM2) are shown in orange, and the extracellular domain is shown in green. (**B**) The exonic structure of the full-length *P2RX7* messenger ribonucleic acid (mRNA) (exons 1 to 13) that encodes the 595 amino acid P2X7A subunit. The amino acid exon boundaries are numbered above the exonic structure (i.e., exon 1 encodes amino acids 1 to 42 of the P2X7R protein, exon 2 encodes amino acids 43 to 99, and so on). The corresponding P2X7A subunit regions that are translated from the mRNA are illustrated below the exonic structure with the colours aligning to the regions shown in (**A**). (**C**) Domains of interest located within the carboxy terminal of the P2X7A subunit including the C rich domain, actin filament binding domain, SH3 binding domain, TNFR1 death domain, trafficking domain, and LPS domain. Structural model produced using PyMOL software version 2.5.2 [23].

**Figure 4 ijms-23-08174-f004:**
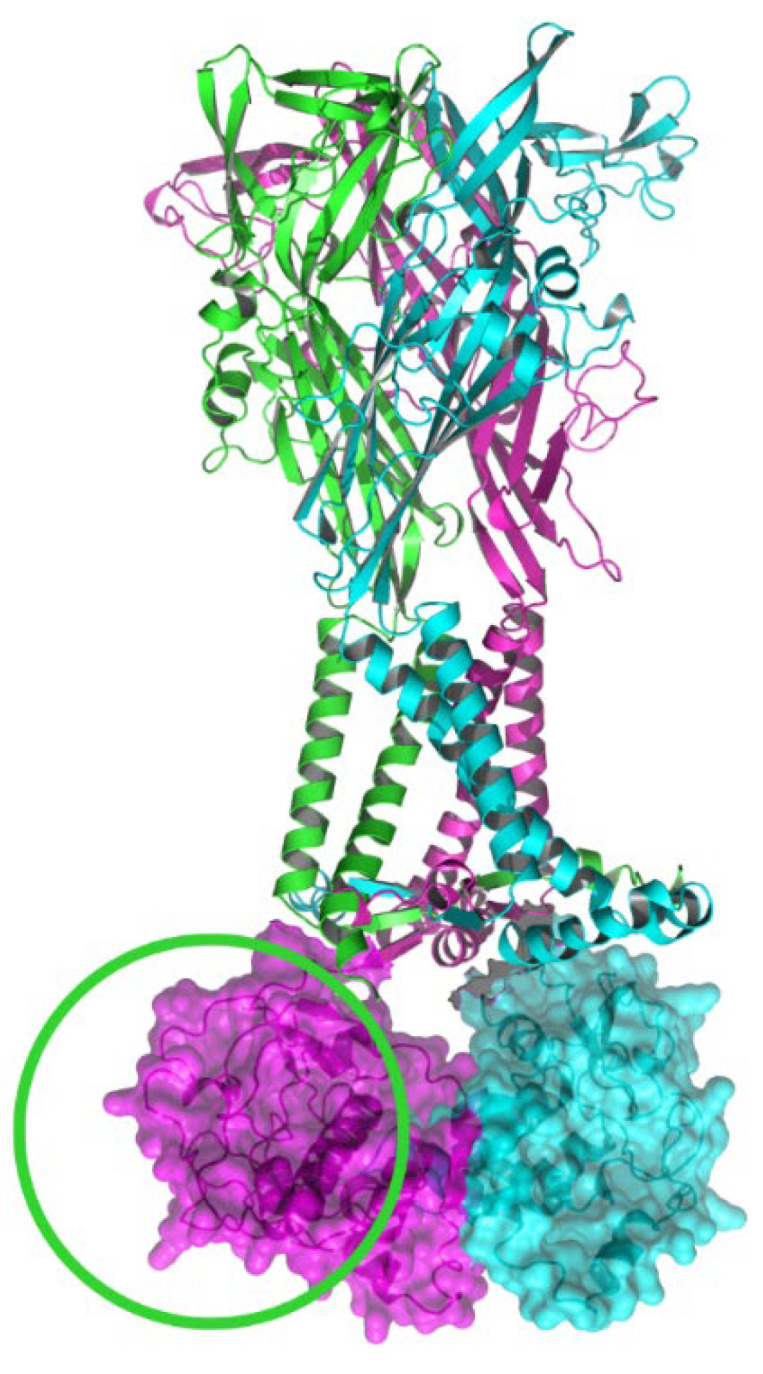
Ribbon representation of the heterotrimeric (P2X7A)_2_/P2X7B receptor. The P2X7B subunit is shown in green, lacking the carboxy terminal as highlighted by the green circle, while the P2X7A subunits are shown in pink and blue. Molecular surface representations of the P2X7A carboxy termini are shown in pink and blue. The structural model has been produced using PyMOL software version 2.5.2 [23].

**Table 1 ijms-23-08174-t001:** Alignment of NCBI and Ensembl transcript isoforms to the commonly used alphabetised naming convention for P2X7R transcripts. Eight protein isoforms are listed on UniProtKB (Q95572-1 to -8) corresponding to the commonly used nomenclature (A–H) as well as four computationally mapped potential isoform sequences. Underlined amino acid numbers indicate sequences translated from the primary start codon. All other numbers refer to P2X7-like proteins that would be translated if alternative start codons were used. Several transcript isoforms are missing in the alphabetised nomenclature: P2X7I has been described as a transcript isoform [13] but has no transcript accession number in NCBI or Ensembl. P2X7I relates to the mutation at the intron 1 donor splice site rs35933842 [80]; P2X7K is a variant containing an alternative intracellular amino terminal and first transmembrane domain encoded by a novel exon 1 in the rodent *P2rx7* gene [81]. A human equivalent has not been identified; P2X7M has not been assigned to any transcript but could be assigned to ∆E2/Predicted isoform X2. ^a^ Transcripts AK225163.1, AK290405.1, BC011913 also match P2X7A; ^b^ Transcript BC007679.2 matches P2X7D; ^c^ This sequence of amino acids matches Q99572-3 exactly. ^d^ This amino acid sequence matches Q99572-6 exactly; ^e^ Described by Sun et al. [78]; ^f^ These sequences are the same; ^g^ Transcript BC121158.1 matches Variant 4; ^h^ Transcript AK090866.1 matches Variant 7; ^i^ These amino acid sequences are the same; ^j^ This transcript (520 bp) aligns exactly to bases 54–573 of Predicted Variant X3; ^k^ This transcript (546 bp) aligns partially with P2X7-213 and Predicted Variant X3.

NCBI				ENSEMBL		UniPRotKB	Possible P2X7-Like Proteins with at Least 1 TMD	
Common mRNA Isoform Name	Accession Number	Other Transcripts with Corresponding Exonic Structure		Transcript Name	Accession Number	Accession Number	Number of Amino Acids	Predicted Molecular Weight (kDa)
		Transcript Name	Accession Number					
A		P2X7 receptor	Y09561	P2X7-202	ENST00000328963.10	Q99572-1	595	68.6
	GQ1801221	Variant 1	NM_002562.5 ^a^					
B	AY847298.1	Variant 5	NR_033951.2	P2X7-203	ENST00000535250.5	Q99572-2	364	41.8
C	AY847299.1	Variant 6	NR_033952.2	P2X7-212	ENST00000541716.5	Q99572-3	128	14.7
							113	13.4
D	AY847300.1	Variant 8	NR_033954.2 ^b^	P2X7-201	ENST0000261826.10	J3KN30	149	17.1
						Q99572-4	425	49.2
E	AY847301.1			P2X7-204	ENST00000535600.2	Q99572-5	275	31.3
F	AY847302.1	Variant 10	NR_033956.2	P2X7-210	ENST00000541022.5		128 ^c^	14.7
						Q99572-6	306	35.5
G	AY847303.1	Variant 2	NR_033948.2	P2X7-208	ENST00000539606.5	F5H2X6	127	14.7
						Q99572-7	274	31.4
H	AY847304.1	Variant 3	NR_033949.2	P2X7-207	ENST00000538011.5	F5H2X6	127	14.7
						Q99572-8	505	58.2
J	DQ399293.1	Variant 9	NR_033955.2	P2X7-211	ENST00000541564.5	F5H8E7	258	29.3
							306 ^d^	35.5
L	MK465687.1	Predicted variant X1	XM_047428912.1				506	58.0
		Predicted variant X2	XM_011538419.4				491	56.7
		∆E2 ^e^					-	-
N	MK465688.1						207	23.5
							344 ^f^	40.1
O	MK465689.1						213	24.3
							272	31.9
P	MK465690.1						258	29.3
Q	MK465691.1						128 ^c^	14.7
							306 ^d^	35.6
		Variant 4	NR_033950.2 ^g^	P2X7-209	ENST00000539695.5		148	17
		Variant 7	NR_033953.2 ^h^	P2X7-206	ENST00000537312.5	F5H237	47 ^i^	5.5
							260	29.9
				P2X7-213 ^j^	ENST00000545434.5	F5H237	47 ^i^	5.5
				P2X7-205 ^k^	ENST00000535928.5	F5H237	47 ^i^	5.5
		Predicted Variant X3	XM_017019367.3				47 ^i^	5.5
							344 ^f^	40.2

## Data Availability

Not applicable.

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
