# Peer review of "Alternatively Spliced Isoforms of the P2X7 Receptor: Structure, Function and Disease Associations"

_ijms, 2022, doi:10.3390/ijms23158174_

Round 1

Reviewer 1 Report

Line 70 ff: Ref 29 provides counterevidence that pore dilation is a myth.  There is strong experimental evidence that the so-called pore dilatation deduced from Vrev shifts is an artifact resulting from profound changes in the concentrations of intracellular ions associated with prolonged P2XR activation (Li M, Toombes GE, Silberberg SD, Swartz KJ Physical basis of apparent pore dilation of ATP-activated P2X receptor channels. Nat Neurosci 18(11):1577–1583, 2015). When ion concentration gradients are tightly controlled as possible in single channel recordings, no such pore dilatation is observed (Riedel et al 2007; Pippel et al 2017). This statement concerns several papers cited in this work

Author Response

ijms-1820791

Alternatively Spliced Isoforms of the P2X7 Receptor: Structure, Function and Disease Associations

Reviewer 1

Line 70 ff: Ref 29 provides counterevidence that pore dilation is a myth.  There is strong experimental evidence that the so-called pore dilatation deduced from Vrev shifts is an artifact resulting from profound changes in the concentrations of intracellular ions associated with prolonged P2XR activation (Li M, Toombes GE, Silberberg SD, Swartz KJ Physical basis of apparent pore dilation of ATP-activated P2X receptor channels. Nat Neurosci 18(11):1577–1583, 2015). When ion concentration gradients are tightly controlled as possible in single channel recordings, no such pore dilatation is observed (Riedel et al 2007; Pippel et al 2017). This statement concerns several papers cited in this work

Authors’ response:

The authors agree with Reviewer 1. Counter evidence to the existence of a pore has now been included. The following has been added to the manuscript and Pippel at al. is now discussed in this section:

However, patch-clamp electrophysiology studies performed by Riedel et al. have shown that single channel kinetics and permeation properties do not change during prolonged receptor activation, challenging the idea of pore dilation [34]. Li et al. found that the characteristic shift in equilibrium, or reversal, potential, found with prolonged P2X7R activation resulted from time-dependent alterations in the concentration of intracellular ions rather than the opening of a larger pore [35]. Furthermore, substituted cysteine accessibility mutagenesis experiments and single channel studies have provided no evidence of pore formation following prolonged ATP activation [36]. Consequently, the existence of other pathways has been postulated to explain permeability to relatively large cations such as NMDG+ and ethidium+ [37-39].

References:

  1. Riedel, T., G. Schmalzing, and F. Markwardt, Influence of extracellular monovalent cations on pore and gating properties of P2X7 receptor-operated single-channel currents. Biophys J, 2007. 93(3): p. 846-58.
  2. Li, M., et al., Physical basis of apparent pore dilation of ATP-activated P2X receptor channels. Nat Neurosci, 2015. 18(11): p. 1577-83.
  3. Pippel, A., et al., Localization of the gate and selectivity filter of the full-length P2X7 receptor. Proc Natl Acad Sci U S A, 2017. 114(11): p. E2156-e2165.
  4. Browne, L.E., et al., P2X7 receptor channels allow direct permeation of nanometer-sized dyes. J Neurosci, 2013. 33(8): p. 3557-66.
  5. Chaumont, S. and B.S. Khakh, Patch-clamp coordinated spectroscopy shows P2X2 receptor permeability dynamics require cytosolic domain rearrangements but not Panx-1 channels. Proc Natl Acad Sci U S A, 2008. 105(33): p. 12063-8.
  6. Pelegrin, P. and A. Surprenant, Pannexin-1 mediates large pore formation and interleukin-1beta release by the ATP-gated P2X7 receptor. Embo j, 2006. 25(21): p. 5071-82.

Reviewer 2 Report

The review manuscript (ijms-1820791) entitled “Alternatively Spliced Isoforms of the P2X7 Receptor: Structure, Function and Disease Associations” by Dr. De Salis (Faculty of Medicine and Health, Sydney Pharmacy School, The University of Sydney, S114, Pharmacy and Bank Building, A15, NSW, 2006, Australia)
Provides a detailed overview of the in the structure and function of alternatively spliced P2X7R isoforms and their associations with different cancer types and human diseases and potential role in modulating the inflammatory response.

The manuscript provides lot of details on P2X7R function, isoforms, disease implication and clinical application. It will increase our knowledge behind the role of P2X7R in a large variety of physiological and pathological processes. The topic is quite important, as the receptor can be considered an important target for tumor therapy. I believe that the work will therefore have an adequate impact in these fields. The text is in general interesting, well written, clear and easy to follow. The majority of references in the field have been included (please see below for additional comments)

My final recommendation is a minor revision.

I have various, minor, comments for improving the manuscript:

I suggest including a brief overview on P2X receptor family in the first paragraph

For completeness, i suggest providing a brief description of the P2X7R general function in the introduction  PMID: 33987781

Lines 39-40 P2x7R activity mainly depends on ligand concentrations. Low ATP concentrations seems to limit the non-selective entry of Ca+2 and Na+ through the channel, while but continuous stimulation at high concentrations of ATP leads to the formation of a non-selective pore wich allow the passage of molecules up to 900 Da (PMID: 21983632)

For completeness, this recent detailed review on P2X7R and its role in inflammation and cancer should be included https://doi.org/10.3390/cancers14051116

In figures 1, 2, 4 the graphic tool used for the molecule representation should be mentioned

Lines 76-77 citations?

The receptor C-terminus is subjected to palmitoylation to regulates P2X7R activities and trafficking PMID: 28920575

308 This subhead title should be modified in “Role of P2X7R isoforms in cancer “ which is more informative

Line 319 a couple of introductive sentences on lung carcinoma would be helpful for the reader

Line 360 better HPV16 and to a lesser extent, HPV18 PMID: 31500479

Line 477 P2X7R and Pannexin-1 are able to complex with P2X4R upon NLRP3 inflammasome activation PMID: 28266268 PMID: 23936165 this information should be included

Author Response

ijms-1820791

Alternatively Spliced Isoforms of the P2X7 Receptor: Structure, Function and Disease Associations

Reviewer 2

The review manuscript (ijms-1820791) entitled “Alternatively Spliced Isoforms of the P2X7 Receptor: Structure, Function and Disease Associations” by Dr. De Salis (Faculty of Medicine and Health, Sydney Pharmacy School, The University of Sydney, S114, Pharmacy and Bank Building, A15, NSW, 2006, Australia)

Provides a detailed overview of the in the structure and function of alternatively spliced P2X7R isoforms and their associations with different cancer types and human diseases and potential role in modulating the inflammatory response.

The manuscript provides lot of details on P2X7R function, isoforms, disease implication and clinical application. It will increase our knowledge behind the role of P2X7R in a large variety of physiological and pathological processes. The topic is quite important, as the receptor can be considered an important target for tumor therapy. I believe that the work will therefore have an adequate impact in these fields. The text is in general interesting, well written, clear and easy to follow. The majority of references in the field have been included (please see below for additional comments)

My final recommendation is a minor revision.

Authors’ response:

I have various, minor, comments for improving the manuscript:

I suggest including a brief overview on P2X receptor family in the first paragraph.

Now included:

P2X receptors are an ancient family of proteins that are expressed in both primitive and advanced life-forms, including protozoa, algae, flatworms, fish, birds, and mammals [1-5]. They are purinergic membrane receptors that assemble as cation channels after binding of extracellular adenosine 5’-triphosphate (ATP) [1]. P2X receptors are further classified into seven subtypes, P2X1 to P2X7, that are each translated from different genes. P2X7 receptors (P2X7R), encoded by the gene P2RX7, are unique from the other subtypes in both their structure and function [2, 6].

For completeness, i suggest providing a brief description of the P2X7R general function in the introduction. PMID: 33987781. Lines 39-40 P2x7R activity mainly depends on ligand concentrations. Low ATP concentrations seems to limit the non-selective entry of Ca+2 and Na+ through the channel, while but continuous stimulation at high concentrations of ATP leads to the formation of a non-selective pore which allow the passage of molecules up to 900 Da (PMID: 21983632)

Added references and modified introduction as below:

A unique property of P2X7Rs is the modulation of their activity based on ATP concentration. At low ATP concentrations the receptor forms a cation channel. At high ATP concentrations the receptor forms a non-selective pore that allows molecules up to 900 Da to pass through the membrane, consequently inducing cell apoptosis [7]. Many other cellular responses are triggered when ATP, binds including cytokine secretion, shedding of cell surface molecules, cell proliferation, and attenuation of P2X7R-dependent phagocytosis [8-11].

For completeness, this recent detailed review on P2X7R and its role in inflammation and cancer should be included https://doi.org/10.3390/cancers14051116

Added to the Introduction:

In addition, the reader is referred to other recent reviews on the P2X7R [11, 12].

In figures 1, 2, 4 the graphic tool used for the molecule representation should be mentioned

Added: Structural model produced using PyMOL software [13].

Lines 76-77 citations?

Added Rotondo et al., 2022.

The receptor C-terminus is subjected to palmitoylation to regulates P2X7R activities and trafficking PMID: 28920575

Reference added.

308 This subhead title should be modified in “Role of P2X7R isoforms in cancer “ which is more informative

Changed.

Line 319 a couple of introductive sentences on lung carcinoma would be helpful for the reader

Added: Lung cancers are divided into small-cell carcinoma and non-small-cell carcinoma based on histology, prognosis, and response to specific therapies [14]. Adenocarcinoma is the most common type of non-small-cell-carcinoma and is comprised of malignant epithelial cells with glandular differentiation [14]

Line 360 better HPV16 and to a lesser extent, HPV18 PMID: 31500479

Modified and reference added:

Cervical cancer is one of the most common cancers diagnosed in women [15], and virtually all cases are caused by infection with human papillomavirus (HPV) [16], and screening for HPV subtype 16 oncoprotein has a higher sensitivity than cytological testing [17].

Line 477 P2X7R and Pannexin-1 are able to complex with P2X4R upon NLRP3 inflammasome activation PMID: 28266268 PMID: 23936165 this information should be included

Added:

A less established mechanism is the activation of the NLRP3 inflammasome through the production of reactive oxygen species that is stimulated by a P2X4/P2X7/pannexin-1 complex. Hung et al. found that eliminating any component of the complex abolished NLRP3 inflammasome   suggesting that alterations to the complex would influence the immune response [18, 19].

References (note the references will be updated in the manuscript)

  1. Chessell, I.P., et al., Cloning and functional characterisation of the mouse P2X7 receptor. FEBS Lett, 1998. 439(1-2): p. 26-30.
  2. Agboh, K.C., et al., Functional characterization of a P2X receptor from Schistosoma mansoni. J Biol Chem, 2004. 279(40): p. 41650-7.
  3. Bo, X., R. Schoepfer, and G. Burnstock, Molecular cloning and characterization of a novel ATP P2X receptor subtype from embryonic chick skeletal muscle. J Biol Chem, 2000. 275(19): p. 14401-7.
  4. Boue-Grabot, E., M.A. Akimenko, and P. Seguela, Unique functional properties of a sensory neuronal P2X ATP-gated channel from zebrafish. J Neurochem, 2000. 75(4): p. 1600-7.
  5. Fountain, S.J., et al., An intracellular P2X receptor required for osmoregulation in Dictyostelium discoideum. Nature, 2007. 448(7150): p. 200-3.
  6. Skarratt, K.K., et al., A P2RX7 single nucleotide polymorphism haplotype promotes exon 7 and 8 skipping and disrupts receptor function. FASEB J, 2020. 34(3): p. 3884-3901.
  7. Notomi, S., et al., Critical involvement of extracellular ATP acting on P2RX7 purinergic receptors in photoreceptor cell death. Am J Pathol, 2011. 179(6): p. 2798-809.
  8. Sluyter, R. and L. Stokes, Significance of P2X7 receptor variants to human health and disease. Recent Pat DNA Gene Seq, 2011. 5(1): p. 41-54.
  9. Wiley, J.S., et al., The human P2X7 receptor and its role in innate immunity. Tissue Antigens, 2011. 78(5): p. 321-32.
  10. Gu, B.J. and J.S. Wiley, P2X7 as a scavenger receptor for innate phagocytosis in the brain. Br J Pharmacol, 2018. 175(22): p. 4195-4208.
  11. Jiang, L.H., et al., Structural basis for the functional properties of the P2X7 receptor for extracellular ATP. Purinergic Signal, 2021. 17(3): p. 331-344.
  12. Rotondo, J.C., et al., The Role of Purinergic P2X7 Receptor in Inflammation and Cancer: Novel Molecular Insights and Clinical Applications. Cancers, 2022. 14(5): p. 1116.
  13. L., S. and D. W., PyMOL [Internet]. 2020: p. http://www.pymol.org/pymol.
  14. Rodriguez-Canales, J., E. Parra-Cuentas, and Wistuba, II, Diagnosis and Molecular Classification of Lung Cancer. Cancer Treat Res, 2016. 170: p. 25-46.
  15. Bray, F., et al., Global cancer statistics 2018: GLOBOCAN estimates of incidence and mortality worldwide for 36 cancers in 185 countries. CA Cancer J Clin, 2018. 68(6): p. 394-424.
  16. Burd, E.M., Human papillomavirus and cervical cancer. Clin Microbiol Rev, 2003. 16(1): p. 1-17.
  17. Okunade, K.S., Human papillomavirus and cervical cancer. J Obstet Gynaecol, 2020. 40(5): p. 602-608.
  18. Hung, S.C., et al., P2X4 assembles with P2X7 and pannexin-1 in gingival epithelial cells and modulates ATP-induced reactive oxygen species production and inflammasome activation. PLoS One, 2013. 8(7): p. e70210.
  19. Orioli, E., et al., P2X7 Receptor Orchestrates Multiple Signalling Pathways Triggering Inflammation, Autophagy and Metabolic/Trophic Responses. Curr Med Chem, 2017. 24(21): p. 2261-2275.
